# Establishment of the First National Standard for Neutralizing Antibodies against SARS-CoV-2 XBB Variants

**DOI:** 10.3390/v16040554

**Published:** 2024-04-01

**Authors:** Xuanxuan Zhang, Lidong Guan, Na Li, Ying Wang, Lu Li, Mingchen Liu, Qian He, Jiansheng Lu, Haiyuan Zeng, Shan Yu, Xinyi Guo, Jiali Gong, Jing Li, Fan Gao, Xing Wu, Si Chen, Qian Wang, Zhongfang Wang, Weijin Huang, Qunying Mao, Zhenglun Liang, Miao Xu

**Affiliations:** 1Institute of Biological Products, National Institutes for Food and Drug Control, NHC Key Laboratory of Research on Quality and Standardization of Biotech Products, State Key Laboratory of Drug Regulatory Science, Beijing 102629, China; zhangxx_01@163.com (X.Z.); guanld@nifdc.org.cn (L.G.); 13717807586@163.com (Y.W.); 18701663972@163.com (L.L.); liumingchen@nifdc.org.cn (M.L.); hq5740@126.com (Q.H.); gaofan@nifdc.org.cn (F.G.); eastarwx@163.com (X.W.); huangweijin@nifdc.org.cn (W.H.); 2Beijing Minhai Biotechnology Co., Ltd., Beijing 102600, China; lln0809@163.com; 3Yunnan Institute for Food and Drug Control, Kunming 650106, China; lujiansheng183@126.com (J.L.); zenghaiyuan2024@163.com (H.Z.); 4Jiangsu Institute for Food and Drug Control, Nanjing 210019, China; yushan@jsifdc.org.cn; 5Hualan Biological Engineering Chongqing Co., Ltd., Chongqing 408107, China; xiyi_98@126.com; 6China Resources Boya Bio-Pharmaceutical Group Co., Ltd., Fuzhou 344000, China; 15079400247@163.com; 7Beijing Kexing Zhongwei Biotechnology Co., Ltd., Beijing 102600, China; lijing@sinovac.com; 8Drug and Vaccine Research Center, Guangzhou National Laboratory, Guangzhou 510535, China; chen_si@gzlab.ac.cn (S.C.); wang_qian@gzlab.ac.cn (Q.W.); wangzhongfang@gird.cn (Z.W.)

**Keywords:** SARS-CoV-2, XBB variants, neutralizing antibody, standard material, standard, reference reagent

## Abstract

Neutralizing antibodies (NtAbs) against severe acute respiratory syndrome coronavirus-2 (SARS-CoV-2) are indicators of vaccine efficacy that enable immunity surveillance. However, the rapid mutation of SARS-CoV-2 variants prevents the timely establishment of standards required for effective XBB vaccine evaluation. Therefore, we prepared four candidate standards (No. 11, No. 44, No. 22, and No. 33) using plasma, purified immunoglobulin, and a broad-spectrum neutralizing monoclonal antibody. Collaborative calibration was conducted across nine Chinese laboratories using neutralization methods against 11 strains containing the XBB and BA.2.86 sublineages. This study demonstrated the reduced neutralization potency of the first International Standard antibodies to SARS-CoV-2 variants of concern against XBB variants. No. 44 displayed broad-spectrum neutralizing activity against XBB sublineages, effectively reduced interlaboratory variability for nearly all XBB variants, and effectively minimized the geometric mean titer (GMT) difference between the live and pseudotyped virus. No. 22 showed a broader spectrum and higher neutralizing activity against all strains but failed to reduce interlaboratory variability. Thus, No. 44 was approved as a National Standard for NtAbs against XBB variants, providing a unified NtAb measurement standard for XBB variants for the first time. Moreover, No. 22 was approved as a national reference reagent for NtAbs against SARS-CoV-2, offering a broad-spectrum activity reference for current and potentially emerging variants.

## 1. Introduction

The coronavirus disease (COVID-19) pandemic has had an unprecedented impact worldwide. As of December 2023, nearly 770 million severe acute respiratory syndrome coronavirus-2 (SARS-CoV-2) cases and over 6.9 million related deaths have been reported globally, severely affecting global public health and socioeconomic structures [1,2,3,4]. Hundreds of companies have developed vaccines based on five major technological approaches to effectively control the pandemic. Inactivated, mRNA, adenovirus vector, attenuated influenza virus vector, and recombinant protein vaccines have been approved for marketing or emergency use, making substantial contributions to global pandemic prevention and control [5,6].

Neutralizing antibodies (NtAbs) against SARS-CoV-2 are crucial indicators of the immunogenicity of COVID-19 vaccines, the efficacy of antibody-based therapeutics, and for conducting seroepidemiological surveys [7,8,9]. The standards for NtAbs act as a globally unified metrological standard for coordinating SARS-CoV-2 NtAb detection. The World Health Organization (WHO) has established the first and second International Standards (IS) for anti-SARS-CoV-2 immunoglobulin (Lot: 20/136, 21/340) [10,11]. In addition, China has established the first and second National Standards for neutralizing antibodies against SARS-CoV-2 (Lot: 280034-202001, 280034-202102) [12,13]. These standards provide a foundation for the accurate and comparable detection of SARS-CoV-2 NtAbs, thereby facilitating the development and application of COVID-19 vaccines and antibody-based drugs worldwide [14]. However, the rapid mutation of SARS-CoV-2 has made it challenging for these standards to meet the detection requirements for variants of concern (VOC). Consequently, the WHO developed the “1st International Standard 2022 Antibodies to SARS-CoV-2 variants of concern 21/338” (1st VOC IS) suitable for detecting NtAbs against early omicron variants in December 2022 [10].

Nonetheless, the XBB variants, characterized by high transmissibility and immune evasion capabilities, have become prevalent globally [15,16]. In May 2023, WHO recommended a vaccination strategy utilizing a monovalent XBB.1 lineage (such as XBB.1.5) to enhance immune responses to circulating SARS-CoV-2 variants [17]. On 13 December 2023, WHO reiterated its recommendation for using monovalent XBB.1.5 as the antigen component in COVID-19 vaccines, owing to the breadth in immune responses elicited by monovalent XBB.1.5 vaccines against circulating variants [18]. Pfizer, Moderna, and Novavax have updated their marketed vaccines with XBB.1.5 antigen, and China has approved six COVID-19 vaccines containing XBB descendent lineages for emergency use [19,20,21,22]. There is an urgent need to establish corresponding NtAb standards to provide a unified value for evaluating the immunogenicity of vaccines containing XBB variant components and conducting seroepidemiological surveys. Therefore, this study aimed to develop a new generation of standards with broad-spectrum neutralizing activity using three types of materials: plasma, purified immunoglobulins, and a broad-spectrum monoclonal antibody (SA55) [23]. To our knowledge, this study is the first to establish a National Standard for SARS-CoV-2 NtAbs against XBB variants.

## 2. Materials and Methods

In December 2022, the National Institute for Food and Drug Control (NIFDC) in China sought to develop a new generation of standards with broad-spectrum neutralizing activity according to the requirements of the “WHO manual for the preparation of reference materials for use as secondary standards in antibody testing” [24]. The institute organized collaborative calibration studies with nine laboratories in China for NtAb detection against 11 strains of SARS-CoV-2: WT, as well as BA.5 (BA.5, BF.7), XBB (XBB.1.5, XBB.1.9, XBB.1.16, XBB.2.3, EG.5 HV.1), and BA.2.86 (BA.2.86, JN.1) sublineages. This paper presents a collaborative calibration study of the four candidate standards.

### 2.1. Materials and Ethics Statement

Six plasma samples collected from COVID-19 convalescent and vaccinated donors between December 2022 and January 2023 were provided by the China Resources Boya Bio-pharmaceutical Group Co., Ltd. (Fuzhou, China) and Hualan Biological Engineering Chongqing Co., Ltd (Chongqing, China). (Appendix A). One batch each of anti-SARS-CoV-2 immunoglobulin (Lot: 202302002) and a broad-spectrum monoclonal antibody (SA55; Lot: KC202212005A) were provided by Hualan Biological Engineering Inc.(Chongqing, China) and Beijing Kexing Zhongwei Biotechnology Co., Ltd (Beijing, China). All samples showed broad-spectrum neutralizing activity against seven SARS-CoV-2 strains: WT, BA.1, BA.2, BA.5, BQ.1, XBB, and XBB.1.5. Plasma samples collected from healthy adults with negative antibodies against SARS-CoV-2 in 2017 were provided by Bohui Biopharmaceutical (Hebei) Co., Ltd. (Langfang, China); all plasma samples were negative for HBsAg, HIV-1/HIV-2, HCV, and syphilis antibodies. All donors provided informed consent for the use of their samples.

### 2.2. Production and Evaluation of the Candidates

Candidate 1 was a frozen preparation of a pool of plasma obtained from three SARS-CoV-2-vaccinated individuals infected with SARS-CoV-2 BA.5 or BF.7 isolates in December 2022. Samples were subjected to heat inactivation for 30 min at 56 °C, followed by defibrination. The pooled plasma was aseptically aliquoted in glass DIN ampoules, each containing 0.2 mL. These were subsequently sealed and cryopreserved at −35 °C.

Candidate 2 was a frozen preparation of a mixture of SA55 and heat-inactivated plasma obtained from three SARS-CoV-2-vaccinated individuals infected with SARS-CoV-2 BA.5 or BF.7 isolates in December 2022. The mixture was aseptically aliquoted in glass DIN ampoules, each containing 0.2 mL, which were then sealed and cryopreserved at −35 °C.

Candidate 3 was a frozen preparation of a mixture of SA55 and heat-inactivated plasma collected from a healthy donor with a negative antibody against SARS-CoV-2 in 2017. The mixed plasma was aseptically aliquoted in glass DIN ampoules, each containing 0.2 mL. These were then sealed and cryopreserved at −35 °C.

Candidate 4 was a frozen preparation of anti-SARS-CoV-2 immunoglobulin prepared according to the immunoglobulin manufacturing process (lot: 202302002). It was aseptically aliquoted in glass DIN ampoules, each containing 0.2 mL. It was subsequently lyophilized, sealed, and cryopreserved at −35 °C.

### 2.3. Collaborative Calibration Study

#### 2.3.1. Samples and Virus

The collaborative study consisted of 10 samples. Sample 10 was the 1st International Standard (IS) 2022 Antibodies to SARS-CoV-2 variants of concern (VOC; coded 21/338) and was purchased from NIBSC [8]. Samples 11, 22, 33, and 44 (Candidates 1–4) were provided by the NIFDC. Sample 55 was a condensed nasal wash collected from individuals who had received inactivated vaccines and were infected with SARS-CoV-2 in May 2023. Samples 66, 77, and 88 were convalescent sera from two donors infected with SARS-CoV-2, provided by the China Resources Boya Bio-pharmaceutical Group Co., Ltd. (Fuzhou, China), with lower or higher titers. Sample 99 was a SARS-CoV-2-negative healthy human serum (Table 1).

In-house live virus (Neut) and pseudotyped virus (PsN) assays were used in this study. The PsN assay used a non-replicative vesicular stomatitis virus (VSV)-based pseudotype virus provided by the NIFDC (Beijing, China) and Beijing Yunling Measuring and Testing Co., Ltd.(Beijing, China).

#### 2.3.2. Participating Laboratories

Nine laboratories with experience in NtAb detection participated in this study: NIFDC (Beijing, China), the Chinese Center for Disease Control and Prevention (Beijing, China), Guangzhou Laboratory (Guangzhou, China), Changping Laboratory (Beijing, China), Beijing Yunling Measuring and Testing Co., Ltd.(Beijing, China), Wuhan Institute of Biological Products Co., Ltd.(Wuhan, China), Sinocelltech Ltd.(Beijing, China), Guangdong Provincial Center for Disease Control and Prevention (Guangzhou, China), and Hubei Provincial Center for Disease Control and Prevention (Wuhan, China). All the laboratories were randomly allocated code numbers between 1 and 9.

#### 2.3.3. Collaborative Calibration Study

The NIFDC organized this collaborative study. Each participating laboratory evaluated the study samples using established methods. The Neut and PsN assays were used to detect antibodies against one or more SARS-CoV-2 strains [25,26,27]. Five laboratories adopted the Neut method to detect NtAbs against BA.5, BF.7, XBB.1.5, XBB.1.9, and XBB.1.16 (Table 2), whereas another five used the PsN method to detect NtAbs against BA.5, BF.7, XBB.1.5, XBB.1.9, XBB.1.16, XBB.2.3, EG.5, HV.1, and JN.1 (Table 2).

The Neut assay is a cytopathic effect-based microneutralization assay [27]. The PsN method was performed as previously described [26]. Participating laboratories performed three independent assays for each challenged strain on different days. At least eight dilutions were used for each assay and for each sample, and at least four wells were used for each parallel dilution.

### 2.4. Statistical Methods

Raw data were submitted to the NIFDC. The endpoint titer of each sample was calculated from the 50% inhibitory dilution (ID50) provided by the participating laboratories using NIFDC Biostats software(1.0). To elucidate the coordination ability of the candidate standard, the relative potency of each sample against the WHO IS and each candidate was calculated by taking the endpoint titer ratio of the sample/WHO IS and the sample/each candidate in the same assay. All log-transformed data were analyzed using a probit model. Model fit was assessed using analysis of variance. The variability between laboratories and assays was expressed using geometric coefficients of variation (GCV). The calculation and analysis software used included Microsoft Excel 2016 (Microsoft Corporation, Redmond, WA, USA, 2016), NIFDC Biostat 1.0 (NIFDC, Beijing, China, 2019), and JMP 13 (SAS Institute, Cary, NC, USA, 1989–2016).

## 3. Results

Nine Chinese laboratories with experience in detecting SARS-CoV-2 NtAbs participated in this collaborative calibration study. These included one national vaccine quality control laboratory, one national laboratory, three disease prevention and control agencies, two vaccine manufacturers, one third-party testing organization, and one research institute. All laboratories returned their results as requested. Five laboratories adopted the Neut method using five strains: BA.5, BF.7, XBB.1.5, XBB.1.9, and XBB.1.16. Five laboratories adopted the PsN method to evaluate 11 strains: WT, BA.5, BF.7, XBB.1.5, XBB.1.9, XBB.1.16, XBB.2.3, EG.5, BA.2.86, HV.1, and JN.1 (Table 2). Each laboratory conducted three or more independent and valid tests on ten samples using these strains, completing a total of 137 tests.

All laboratories reported negative results for sample No. 99, resulting in a 100% negative conformity rate. Analysis of the remaining nine samples revealed a substantial anomaly in one result from No. 11 against BF.7 in LB5 with one test being negative and two tests showing titers of 431 and 1448. Therefore, this result was excluded from further analysis. The remaining results were included in subsequent analyses.

### 3.1. Neutralizing Activity of the Candidate Standards

The 1st VOC IS exhibited lower NtAb titers against the latest prevalent strains. Two laboratories using the Neut method reported all four tests against XBB.1.9 and XBB.1.16 as negative; two laboratories using the PsN method reported all nine tests against XBB.1.5, XBB.1.9, XBB.2.3, and BA.2.86 as negative; and all six tests against HV.1 and JN.1 were negative. This suggests that the 1st VOC IS may no longer meet the needs of the most recently prevalent strains, including XBB.1.5, XBB.1.9, XBB.1.16, XBB.2.3, BA.2.86, HV.1, and JN.1. However, Sample Nos. 11, 22, 33, and 44 demonstrated broader-spectrum neutralizing activity against XBB sublineages ranging from XBB.1.5 to HV.1.

When tested using the Neut method, the geometric mean titers (GMTs) of NtAb for No. 11, No. 22, No. 33, and No. 44 against five strains within the BA.5 (BA.5, BF.7) and XBB sublineages (XBB.1.5, XBB.1.9, and XBB.1.16) were 69.1–356.7, 1071.3–2169.8, 782.0–1696.9, and 112.8–857.9, respectively. The GMT for each candidate standard against XBB.1.16 decreased by factors of 4.0, 1.2, 1.1, and 6.9, respectively, compared to BA.5. In contrast, the 1st VOC IS decreased by 8.3 times.

When tested using the PsN method, the GMTs for No. 11, No. 22, No. 33, and No. 44 against nine strains (WT, BA.5 (BA.5, BF.7), XBB sublineages (XBB.1.5, XBB.1.9, XBB.1.16, XBB.2.3, EG.5, HV.1)) were 43–1614, 866–22986, 780–22213, and 95–4428, respectively. For the newly emerged BA.2.86 and JN.1 (BA.2.86 sublineages), the GMTs were <30–65, 3967–21,516, 4738–19,671, and 54–317, respectively. All samples exhibited good broad-spectrum neutralizing activity against XBB sublineages, although GMTs gradually decreased with the evolution of the variant lineage. However, the GMTs decreased by 31.2, 4.1, 4.0, and 42.1 times, respectively, compared to the WT, whereas the 1st VOC IS decreased by 136.1 times for the latest XBB sublineage (HV.1). No. 22 and No. 33 demonstrated exceptional broad-spectrum neutralizing activity against newly emerged variants, with GMTs for BA.2.86 and JN.1 increasing by 6.0, 6.4, 1.1, and 1.5 times, respectively, compared to WT, whereas the 1st VOC IS decreased by 92.8 and 136.1 times for these strains. Strains EG.5, HV.1, BA.2.86, and JN.1 emerged as new variants. These results suggest that Sample Nos. 22 and 33 exhibit excellent broad-spectrum neutralizing activity against newly emerged variants, whereas Nos. 44 and 11 show good neutralizing activity against the XBB lineages, with Sample No. 44 exhibiting higher activity than No. 11 (Figure 1).

### 3.2. Inter-Laboratory Variability

#### 3.2.1. Variability among Laboratories for Candidate Standards

Variability between two or more laboratories was expressed using geometric coefficients of variation (GCV, Table 3 and Table 4). When calculated using the original endpoint, the inter-laboratory variability for all strains tested using the two NtAb detection methods was lower for No. 11 and No. 44 than for No. 22 and No. 33. This was particularly evident in the tests for the XBB sublineages, where No. 11 and No. 44 exhibited lower inter-laboratory variability (Appendix A).

#### 3.2.2. Inter-Laboratory Variability after Normalization

To further assess the impact of normalization using the four candidate standards on interlaboratory variability, relative titers (RT/10, RT/11, RT/22, RT/33, and RT/44) for all samples were calculated using Nos. 10, 11, 22, 33, and 44 as standards. The interlaboratory variability (GCV) was subsequently recalculated.

For the Neut method, Nos. 10, 11, and 44 reduced the inter-laboratory variability for all samples except for No. 55 (a nasal mucosa sample) compared with the original endpoint. The candidate standards were tested for all strains, including XBB.1.5, XBB.1.9, and XBB.1.16. In contrast, Nos. 22 and 33 not only failed to reduce the inter-laboratory variability for most samples but also increased this variance (Table 3).

For the PsN method (Table 4), Nos. 11 and 44 reduced the inter-laboratory variability for nearly all samples of the XBB sublineages despite the inherently smaller variance of this method, performing better than the WHO IS. Similarly, Nos. 22 and 33 exhibited increased variability in most laboratory detection methods.

### 3.3. Inter-Assay Variability

#### 3.3.1. Correlation between Neut and PsN Methods

The relative titers (RT/10, RT/11, RT/22, RT/33, and RT/44) for all samples were calculated to further assess the impact of normalization on the correlation between the two primary global NtAbs detection methods using the candidate standards. Statistical analyses were subsequently conducted to determine the correlation between the two methods before and after normalization. The results are shown in Figure 2 and Appendix A. When using endpoint values, both methods exhibited significant correlation for all variant strains tested, including BA.5, BF.7, XBB.1.5, XBB.1.9, and XBB.1.16 (*p* < 0.001), with correlation coefficients (r values) ranging from 0.9185 to 0.9890. This indicates good correlation between the Neut and PsN methods. After normalization using Nos. 11, 22, 33, and 44, the *p*-values remained <0.001, and the r values ranged from 0.8930 to 0.9892. This suggests that standardization with any of the four candidate standards did not alter the correlation between the Neut and PsN methods. However, the two methods were highly correlated.

#### 3.3.2. Differences in GMT between Detection Methods before and after Normalization

To further explore the differences in GMT between the Neut and PsN methods before and after normalization, the ratios of GMT values obtained using the PsN method to those obtained using the Neut method (Ratio PsN/Live or Live/PsN; Table 5) were calculated using the endpoint titers RT/10, RT/11, RT/22, RT/33, and RT/44. For the BA.5, BF.7, and XBB.1.9 strains, calculating the relative potency using Nos. 10, 11, and 44 reduced the detection differences between the two methods for all samples, with a more pronounced effect than that achieved with Nos. 22 and 33 in minimizing the differences in detection outcomes for polyclonal samples between the two methods.

Evaluation of the XBB.1.5 and XBB.1.16 strains revealed closer original results of the two methods, with ratios ranging between 1.49–5.06 and 1.72–7.31, respectively. Calculating the relative potency using Nos. 10, 11, and 44 also reduced the differences in the detection outcomes between the two methods for all samples. Moreover, the use of RT/11 and RT/44 did not significantly increase the ratio for individual samples in which the differences in the original results were very small (ratio < 2). This suggests that calculating the relative potency of each sample using Nos. 11 and 44 can effectively reduce the differences in detection outcomes between the two methods across all samples.

## 4. Discussion

The ongoing mutation and spread of SARS-CoV-2 poses a major threat to global public health [28]. Mutations in the spike protein, particularly in the receptor-binding domain (RBD), may alter the protein conformation and affect the interaction with the angiotensin converting enzyme 2 (ACE2) receptor, thereby conferring NtAb resistance, immune escape, and high risk of re-infection [29]. Among the variants, XBB.1.5 with a Ser486Pro mutation in RBD, displays the greatest evasion against neutralizing antibodies, endangering the efficacy of COVID-19 vaccines developed based on the ancestral strain of SARS-CoV-2 [30,31]. To improve protection against infection and reduce the transmission of SARS-CoV-2, the Technical Advisory Group on COVID-19 Vaccine Composition (TAG-CO-VAC) recommends the use of a monovalent XBB.1 descendent lineage as the vaccine antigen [17]. To this end, nine COVID-19 vaccines containing XBB sublineages have been marketed or approved for emergency use worldwide. Of these, four are mRNA vaccines, four are recombinant protein vaccines, and one is an inhalable adenovirus vector vaccine. Given the continuous evolution of SARS-CoV-2, TAG-CO-VAC further encourages the clinical evaluation of new vaccine antigen components (lineages derived from XBB and BA.2.86) and the acquisition of immune response results from monovalent XBB.1.5 vaccines across different technological platforms. In addition, this advisory group has made recommendations for enhancing epidemiological and virological surveillance of SARS-CoV-2 to provide a scientific basis for timely updates on COVID-19 vaccine antigen components [18]. The standards for NtAbs against SARS-CoV-2 variants are the norm to ensure the consistency and comparability of clinical evaluations of vaccines against SARS-CoV-2 variants. In this study, the results indicated that the 1st VOC IS had considerably reduced neutralizing antibody potency against seven recently prevalent strains (XBB.1.5, XBB.1.9, XBB.1.16, XBB.2.3, BA.2.86, HV.1, and JN.1), even turning negative in some cases. Therefore, it is necessary to establish standards for the detection of NtAbs against SARS-CoV-2 variants, specifically against XBB sublineages, to meet the current needs for the clinical evaluation of vaccines targeting XBB and other variants.

As metrological standards for biological activity, biological standard substances often use raw materials that are homogenous or interchangeable with the test samples. For instance, antibody standards are typically developed using convalescent and/or post-vaccination plasma or human immunoglobulins, which possess sufficient activity and can effectively harmonize the detection methods [24]. However, the development of COVID-19 standards using convalescent human plasma or immunoglobulins has been constantly restricted by the rapid mutation and increasing immune evasion of SARS-CoV-2. Conducting standard substance research for continuously mutating pathogens, such as SARS-CoV-2, presents another critical challenge for researchers worldwide. In addition to using plasma and immunoglobulins to prepare candidate standards No. 11 and 44, this study also utilized a broad-spectrum anti-SARS-CoV-2 monoclonal antibody (SA55) mixed with convalescent human plasma and anti-SARS-CoV-2-negative plasma separately to prepare Nos. 22 and 33. Nos. 11 and 44 exhibited NtAb GMTs of 43–1614 and 95–4428, respectively, against nine strains (WT, BA.5, BF.7, XBB.1.5, XBB.1.9, XBB.1.16, XBB.2.3, EG.5, and HV.1), showing good neutralizing capability against XBB sublineages. Furthermore, Nos.11 and 44 effectively reduced the detection variability between different laboratories for most serum samples against XBB sublineages, markedly reduced the variability between Neut and PsN methods, and ensured a strong correlation between Neut and PsN results (*p* < 0.001, No. 11 r: 0.9222–0.9892; No. 44 r: 0.9255–0.9890). The results indicate that Nos.11 and 44 demonstrated good method harmonization capabilities for XBB sublineages, including XBB.1.5, meeting the requirements for reference standard development. Interestingly, Nos. 11 and 44 had a weaker reduction capability for mucosal samples (No. 55) than for serum samples, suggesting that there might be differences in the neutralizing characteristics of serum and mucosal antibodies. Approximately 90% of the antibodies in the upper respiratory tract mucosa are sIgA, whereas approximately 80% of those in the blood are IgG. Mucosal sIgA is a dimeric IgA that contains a secretory component that differs from monomeric IgA in the blood [32]. Harold et al. [33] found that the neutralizing activity of sIgA was four, 1.6, and 143 times higher than that of IgA, dIgA, and IgG, respectively. However, whether this is related to the superior protective ability of mucosal vaccines against infections warrants further in-depth investigation.

Nos. 22 and 33 contained a broad-spectrum neutralizing monoclonal antibody, namely, SA55 [23]. This rare monoclonal antibody, obtained by screening the B cells of convalescent patients, can specifically bind to the up-side of the SARS-CoV-2 receptor-binding domain (RBD). It effectively blocks the binding of the SARS-CoV-2 RBD to the ACE2 receptor, demonstrating high efficacy and broad-spectrum neutralizing capability against the currently prevalent SARS-CoV-2 variants, such as XBB.1.5 and EG.5. Nos. 22 and 33, which contain SA55, exhibited exceptional broad-spectrum neutralizing activity against all strains in this study. The GMT values for Nos. 22 and 33 against the recent XBB descendant HV.1 decreased only by 4.1- and 4.0-fold, respectively, compared to the WT, whereas those for the 1st VOC IS decreased by 136.1-fold. The GMT values for No. 22 and No. 33 against the latest prevalent strain of the BA.2.86 sublineages (JN.1) increased by 6.4- and 1.5-fold, respectively, compared to the WT, whereas those of the 1st VOC IS decreased by 136.1-fold. Both candidate standards showed outstanding broad-spectrum neutralizing activity, suggesting potential application in NtAb detection against emerging variants. However, they could not alter the detection variability among different laboratories for the same strains, which may be related to considerable differences in the properties between this monoclonal antibody and polyclonal antibodies derived from clinical samples. This indicates that it may be difficult to achieve homogeneity and a broad spectrum for neutralizing antibody standards for SARS-CoV-2 variants.

According to WHO guidelines, an international standard (IS) is a certified reference standard for biological materials [24]. An IS should be confirmed to possess good biological activity and harmonization capability for detection methods through collaborative calibration. On the other hand, an international reference reagent (IRR) refers to other standard materials that have not undergone extensive collaborative calibration or have been shown through such calibration to be unsuitable as an IS [34]. Consequently, this study established the first National Standard material for neutralizing antibodies against the XBB variant, No. 44 (Lot: 280035-202301, 1000 U/mL). The established standard exhibits broad-spectrum neutralizing activity and good potency against XBB sublineages, creating a unified metric for the detection of NtAbs against XBB variants. No. 22, which demonstrates exceptional broad-spectrum neutralizing activity and higher potency against XBB and BA.2.86 sublineages, but cannot effectively reduce interlaboratory variability, was designated as the national reference reagent for SARS-CoV-2 NtAbs (Lot: 280036-202301). It provides a reference and benchmark for rapidly establishing or evaluating detection methods for NtAbs against emerging variants in response to new mutations. But, for rapidly mutating viruses that evade the immune system, such as SARS-CoV-2, the establishment and application of reference standards, the timing of their initiation, and bridging the metrological relationships between different variant standards require further exploration to develop a more suitable route for the creation of SARS-CoV-2 standards.

## Figures and Tables

**Figure 1 viruses-16-00554-f001:**
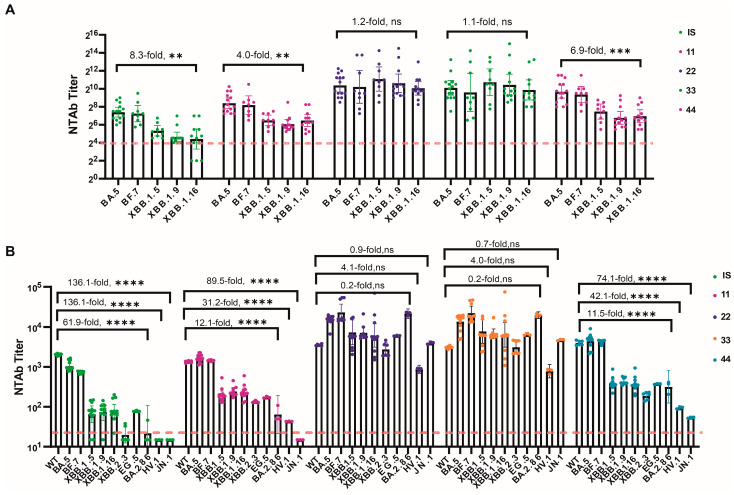
SARS-CoV-2 NtAb titers for all collaborative calibration study samples across all participants. (**A**) NtAb titers against live-virus BA.5, BF.7, XBB.1.5, XBB.1.9, and XBB.1.16; (**B**) NtAbs titers against pseudovirus WT, BA.5, BF.7, XBB.1.5, XBB.1.9, XBB.1.16, XBB.2.3, EG.5, BA.2.86, HV.1, and JN.1. The NtAb titers against live-virus below eight, were defined as four. The NtAb titers against pseudovirus below 30, were defined as 15. Dunnett’s multiple comparison test was performed for comparison, as indicated in the figures; ** *p* < 0.01; *** *p* < 0.001; **** *p* < 0.0001; ns, not significant.

**Figure 2 viruses-16-00554-f002:**
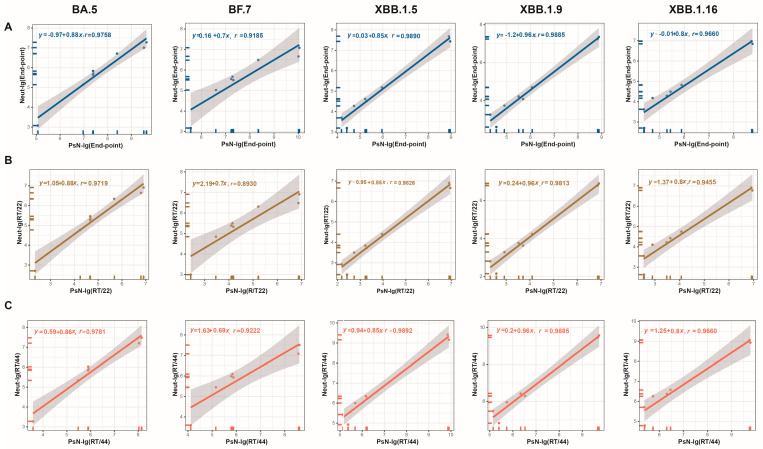
Correlation of endpoint titers and RT/22 and RT/44 between the Neut and PsN methods. (**A**) Endpoint titers-BA.5, BF.7, XBB.1.5, XBB.1.9, and XBB.1.16; (**B**) titers relative to No. 22-BA.5, BF.7, XBB.1.5, XBB.1.9, and XBB.1.16; (**C**) titers relative to No. 44-BA.5, BF.7, XBB.1.5, XBB.1.9, and XBB.1.16. The *p* values of linear fits were all less than 0.001.

**Table 1 viruses-16-00554-t001:** Collaborative study samples.

Sample Code	Description	Formulation
10	1st International Standard 2022 Antibodies to SARS-CoV-2 variants of concern, 21/338	freeze-dry
11	Candidate 1, convalescent plasma, positive	liquid
22	Candidate 2, positive plasma with SA55	liquid
33	Candidate 3, negative plasma with SA55	liquid
44	Candidate 4, human anti-SARS-CoV-2 immunoglobulin	liquid
55	Concentrated nasal wash	liquid
66	Convalescent plasma from one donor, medium neutralizing capacity against XBB	liquid
77	Convalescent plasma from one donor with high NtAb titers against XBB	liquid
88	Convalescent plasma from one donor with low NtAb titers against XBB	liquid
99	Negative plasma	liquid

**Table 2 viruses-16-00554-t002:** The variants used for neutralization assays.

Detection Methods	Lab	Challenge Virus											Summary
		Ancestral	BA.5 Sublineages		XBB Sublineages						BA.2.86 Sublineages		
		WT	BA.5	BF.7	XBB.1.5	XBB.1.9	XBB.1.16	XBB.2.3	EG.5	HV.1	BA2.86	JN.1	
Neut assay	LB1	/		√	√	/	√	/	/	/	/	/	4
	LB2	/	√	/	/	√	/	/	/	/	/	/	2
	LB3	/	/	/	/	√	√	/	/	/	/	/	2
	LB4	/	√	√	√	√	√	/	/	/	/	/	5
	LB5	/	√	√	√	√	√	/	/	/	/	/	5
PsN assay	LB2	/	√	/	/	√	/	/	/	/	/	/	2
	LB6	/	√	/	√	/	√	/	/	/	/	/	3
	LB7	√	√	√	√	√	√	√	√	/	/	/	8
	LB8	√	√	√	√	√	√	√	/	√	√	√	10
	LB9	/	√	√	√	/	√	/	/	/	/	/	4

**Table 3 viruses-16-00554-t003:** Geometric coefficients of variation (%, GCV) of Neut method endpoint titers and relative titers against Nos. 10, 11, 22, 33, and 44 across all participants.

Challenge Virus	GCV, %	Sample Code								
		10	11	22	33	44	55	66	77	88
BA5	Endpoint	95.4	112.4	152.1	141.5	149.2	94.6	115.6	84.1	103.0
	RT/10	/	10.7	71.1	113.1	33.0	25.5	14.0	13.0	4.5
	RT/11	10.7	/	61.1	108.6	20.7	31.9	3.7	22.1	7.4
	RT/22	136.9	313.9	/	146.3	92.9	189.2	118.0	134.8	127.2
	RT/33	113.1	108.6	42.1	/	101.5	154.7	103.3	104.7	111.4
	RT/44	33.0	20.7	46.9	101.5	/	56.6	17.2	42.3	28.0
BF.7	Endpoint	131.4	148.6	506.8	742.5	132.0	51.7	132.0	96.8	107.8
	RT/10	/	27.8	167.1	270.6	14.3	58.2	33.0	41.6	37.5
	RT/11	27.8		126.8	179.4	32.8	88.4	44.6	79.4	40.3
	RT/22	167.1	126.8		38.9	181.6	322.5	211.2	261.2	243.5
	RT/33	270.6	179.4	38.9	/	289.0	486.2	326.7	397.7	372.6
	RT/44	14.3	32.8	181.6	289.0	/	54.3	16.5	28.6	23.2
XBB.1.5	Endpoint	73.5	69.8	257.8	322.2	128.3	51.9	73.5	73.5	84.3
	RT/10	/	5.9	106.2	143.8	40.3	41.6	10.5	35.7	16.5
	RT/11	5.9	/	111.9	151.6	48.2	45.6	16.5	42.1	12.2
	RT/22	106.2	111.9	/	20.0	70.5	167.1	109.1	135.1	101.4
	RT/33	143.8	151.6	20.0	/	93.9	208.1	144.7	169.2	140.7
	RT/44	40.3	48.2	70.5	93.9	/	59.5	33.0	38.9	54.3
XBB.1.9	Endpoint	93.6	70.9	245.8	299.0	141.1	68.0	65.1	37.2	66.7
	RT/10	/	27.4	99.5	117.0	39.7	30.2	34.7	45.5	23.3
	RT/11	27.4	/	102.8	135.8	41.6	30.1	9.0	26.2	7.7
	RT/22	99.5	102.8	/	33.0	46.5	132.8	110.9	154.5	110.5
	RT/33	117.0	128.0	29.7	/	81.7	173.1	156.7	176.9	137.8
	RT/44	39.7	41.6	46.5	73.2	/	59.2	50.3	78.3	47.1
XBB.1.16	Endpoint	209.2	106.8	123.1	224.8	131.5	80.7	97.6	77.3	230.7
	RT/10	/	95.2	93.9	128.3	101.4	81.0	95.8	88.1	133.5
	RT/11	95.2	/	21.5	98.4	21.0	26.3	12.8	24.4	69.7
	RT/22	93.9	21.5	/	63.6	7.8	36.7	14.6	33.7	77.9
	RT/33	128.3	98.4	63.6	/	66.7	111.2	84.9	107.7	106.4
	RT/44	101.4	21.0	7.8	66.7	/	43.3	18.8	40.1	56.2

Red represents the GCV of endpoint titers. Green represents lower GCV of relative titers than that of endpoint titers. Black represents equal or higher GCV of relative titers than that of endpoint titers.

**Table 4 viruses-16-00554-t004:** Geometric coefficients of variation (%, GCV) of PsN method endpoint titers and relative titers against Nos. 10, 11, 22, 33, and 44 across all participants.

Challenge Virus	GCV, %	Sample Code								
		10	11	22	33	44	55	66	77	88
BA5	Endpoint	27.1	24.3	20.2	61.9	46.6	26.9	46.1	33.2	28.9
	RT/10	/	19.4	40.0	88.4	41.8	25.4	36.9	30.3	31.4
	RT/11	19.4	/	39.6	76.6	24.6	7.4	19.3	9.7	15.8
	RT/22	40.0	39.6	/	40.1	69.1	43.4	65.2	48.2	33.5
	RT/33	88.4	76.6	40.1	/	105.7	78.1	99.4	78.9	56.6
	RT/44	41.8	24.6	69.1	105.7	/	18.6	12.1	18.8	36.5
BF.7	Endpoint	8.6	1.5	104.4	81.4	13.2	45.5	2.6	21.2	8.4
	RT/10	/	10.2	94.7	71.1	19.4	55.5	11.0	30.3	12.1
	RT/11	10.2	/	106.0	83.1	12.8	44.1	1.8	19.9	8.3
	RT/22	94.7	106.0	/	17.2	130.9	195.9	109.5	146.2	92.1
	RT/33	71.1	83.1	17.2	/	104.0	163.8	86.0	119.6	72.1
	RT/44	19.4	12.8	130.9	104.0	/	30.4	10.8	10.7	22.1
XBB.1.5	Endpoint	110.0	37.1	200.8	186.6	35.2	50.2	9.7	19.8	36.7
	RT/10	/	73.4	148.2	142.5	80.3	72.1	109.3	126.6	53.8
	RT/11	73.4	/	122.9	112.7	4.9	15.9	29.1	38.4	23.9
	RT/22	148.2	122.9	/	5.4	123.1	104.4	177.9	190.5	157.9
	RT/33	142.5	112.7	5.4	/	112.7	95.4	164.5	176.1	147.4
	RT/44	80.3	4.9	123.1	112.7	/	18.2	28.2	45.9	23.3
XBB.1.9	Endpoint	73.9	33.0	48.3	26.1	23.6	17.3	20.9	23.0	15.0
	RT/10	/	49.0	37.3	44.0	50.9	80.8	57.8	73.2	62.3
	RT/11	49.0	/	13.0	9.3	7.8	25.5	10.4	17.8	16.1
	RT/22	37.3	13.0	/	18.1	20.2	41.8	24.4	32.8	30.6
	RT/33	44.0	9.3	18.1	/	5.0	25.6	9.7	20.6	13.0
	RT/44	50.9	7.8	20.2	5.0	/	20.1	4.6	14.9	8.8
XBB.1.16	Endpoint	84.2	45.7	196.6	214.4	49.9	44.4	29.5	42.7	26.3
	RT/10	/	30.6	61.2	72.5	23.2	44.3	57.7	43.0	51.2
	RT/11	30.6	/	107.3	119.5	14.2	18.2	28.3	18.4	20.4
	RT/22	61.2	107.3	/	12.3	98.4	125.2	147.6	121.5	140.7
	RT/33	72.5	119.5	12.3	/	111.5	141.6	157.7	130.1	152.7
	RT/44	23.2	14.2	98.4	111.5	/	27.6	32.4	24.8	25.3

Red represents the GCV of endpoint titers. Green represents lower GCV of relative titers than that of endpoint titers. Black represents equal or higher GCV of relative titers than that of endpoint titers.

**Table 5 viruses-16-00554-t005:** The GMT ratios between the PsN and Neut methods.

Variants		Sample Number								
		10	11	22	33	44	55	66	77	88
BA5	Endpoint	5.92	4.53	11.45	12.85	5.16	6.88	5.19	5.43	4.58
	RT/10	/	1.31	1.93	2.17	1.15	1.16	1.14	1.09	1.29
	RT/11	1.31	/	2.53	2.84	1.14	1.52	1.15	1.20	1.01
	RT/22	2.74	5.06	/	1.26	3.14	2.36	3.12	2.98	3.53
	RT/33	2.17	2.84	1.12	/	2.49	1.87	2.48	2.37	2.80
	RT/44	1.15	1.14	2.22	2.49	/	1.33	1.01	1.05	1.13
BF.7	Endpoint	4.83	4.47	19.24	28.40	6.44	10.80	6.13	5.29	4.19
	RT/10	/	1.21	3.99	5.89	1.33	2.24	1.27	1.10	1.15
	RT/11	1.21	/	4.65	7.35	1.62	2.61	1.51	1.37	1.03
	RT/22	3.99	4.65	/	1.48	2.99	1.78	3.14	3.64	4.59
	RT/33	5.89	7.35	1.48	/	4.41	2.63	4.63	5.37	6.77
	RT/44	1.33	1.62	2.99	4.41	/	1.68	1.05	1.22	1.54
XBB.1.5	Endpoint	1.62	2.17	3.40	4.56	2.18	5.06	1.58	1.82	1.49
	RT/10	/	1.34	2.10	2.82	1.35	3.13	1.03	1.12	1.09
	RT/11	1.34	/	1.57	2.11	1.01	2.34	1.37	1.19	1.46
	RT/22	2.10	1.57	/	1.34	1.56	1.49	2.16	1.87	2.29
	RT/33	2.82	2.11	1.34	/	2.09	1.11	2.89	2.51	3.07
	RT/44	1.35	1.01	1.56	2.09	/	2.31	1.46	1.22	1.60
XBB.1.9	Endpoint	3.29	3.54	4.31	4.66	3.83	8.42	3.13	4.93	3.03
	RT/10	/	1.07	1.31	1.41	1.16	2.55	1.05	1.50	1.09
	RT/11	1.07	/	1.22	1.32	1.08	2.38	1.13	1.39	1.17
	RT/22	1.31	1.22	/	1.08	1.13	1.95	1.38	1.14	1.42
	RT/33	1.41	1.23	1.14	/	1.20	1.76	1.45	1.18	1.51
	RT/44	1.16	1.08	1.13	1.22	/	2.20	1.22	1.29	1.26
XBB.1.16	Endpoint	3.67	2.59	5.75	6.79	2.91	7.31	1.72	2.73	1.91
	RT/10	/	1.41	1.57	1.85	1.26	2.00	2.13	1.34	1.92
	RT/11	1.41	/	2.22	2.62	1.12	2.82	1.51	1.05	1.40
	RT/22	1.57	2.22	/	1.18	1.97	1.27	3.34	2.11	3.25
	RT/33	1.85	2.62	1.18	/	2.33	1.08	3.95	2.49	3.77
	RT/44	1.26	1.12	1.97	2.33	/	2.51	1.69	1.07	1.53

Red represents the GMT ratio of endpoint titers between the PsN and Neut methods. Green represents lower GMT ratio of relative titers than that of endpoint titers. Black represents equal or higher GMT ratio of relative titers than that of endpoint titers.

## Data Availability

All data were generated or analyzed, and the materials used in this study are included in this article.

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
