# Peer review of "Establishment of the First National Standard for Neutralizing Antibodies against SARS-CoV-2 XBB Variants"

_viruses, 2024, doi:10.3390/v16040554_

Round 1
Reviewer 1 Report
Comments and Suggestions for Authors
The goal of the manuscript by Zhang and colleagues is to establish a national standard for neutralizing antibodies against SARS-CoV-2 XBB and BA.2.86 variants. Overall this is a straightforward study involving nine different laboratories in China and four different antibody standards (human plasma, immunoglobulin, and a broad-spectrum neutralizing monoclonal antibody) that examined virus neutralization of 11 different variants. The authors used standard virus neutralization tests with live viruses and VSV pseudotyped viruses with the Spike protein of SARS-CoV-2. Overall, this is a well-conceived study with proper controls. The manuscript is also well-written and I only have one major comment and a few minor comments that I believe would strengthen the manuscript.
Major comment:
1. The authors should consider discussing the mutations in the RBD as they may be important in the evolution of the virus and immune evasion.
Minor comments:
1. Line 26: Please define “XBB.”
2. Line 27: should this be purified immunoglobulin?
3. Line 33: Please define “GMT.”
4. Line 66: Please define “VOC IS.”
5. line 389-390: I question whether SARS-CoV-2 continually evades the immune system as the virus is cleared by the host. With that said, the sentence containing “But, for these rapidly mutating and continually immune evasion viruses, such as SARS-CoV-2,” would read better as, “But, for rapidly mutating viruses that evade immune system such as SARS-CoV-2,”
7. line 390: Consider replacing “standard substances” with “standard biologics.”
Comments on the Quality of English LanguageNone
Author Response
Comments 1: The authors should consider discussing the mutations in the RBD as they may be important in the evolution of the virus and immune evasion.
Response: Thanks for your suggestion. Mutations in spike protein, particularly in the receptor-binding domain (RBD), may alter the protein conformation and affect the interaction with the angiotensin con-verting enzyme 2 (ACE2) receptor, thereby conferring NtAb resistance, immune escape, and high risk of re-infection. Among the variants, XBB.1.5 with a Ser486Pro mutation in RBD, displays the greatest evasion against neutralizing antibodies, endangering the efficacy of COVID-19 vaccines developed based on the ancestral strain of SARS-CoV-2.
We have added the discussion about RBD mutations in line 301-306.
Comments 2: Line 26: Please define “XBB.”
Response: “XBB” is a sublineage of Omicron. Since the abstract is word limited and XBB is widely known, we did not define it.
Comments 3: Line 27: should this be purified immunoglobulin?
Response: Thanks for your suggestion. We have corrected the word.
Comments 4: Line 33: Please define “GMT.”
Response: We are very sorry for not define the abbreviation of “GMT”. The meaning of “GMT” is geometric mean titer. We have defined it in line 33.
Comments 5: Line 66: Please define “VOC IS.”
Response: Thank you for pointing this out. In line 66 “1st VOC IS” is the abbreviation of “1st International Standard 2022 Antibodies to SARS-CoV-2 variants of Concern.” We have defined the abbreviation of “IS” and “VOC” in line 58, 64.
Comments 6: line 389-390: I question whether SARS-CoV-2 continually evades the immune system as the virus is cleared by the host. With that said, the sentence containing “But, for these rapidly mutating and continually immune evasion viruses, such as SARS-CoV-2,” would read better as, “But, for rapidly mutating viruses that evade immune system such as SARS-CoV-2,”
Response: We are very sorry for our incorrect writing. The sentence in line 391 has been revised.
Comments 7: line 390: Consider replacing “standard substances” with “standard biologics.”
Response: Thanks for your suggestion. The “standard substances” was replaced with “reference standards” in line 392

Reviewer 2 Report
Comments and Suggestions for Authors
In this manuscript, Zhang et al evaluated the neutralizing activity of NtAbs against several SARs-CoV-2 variants. They produced 4 candidate standards from human plasma, immunoglobulin, and a broad-spectrum neutralizing monoclonal antibody. They found that No.44 (No.22 also) showed the strong neutralizing activity against XBB variants. Overall, the manuscript and experiments were well organized and written. I have some comments on this manuscript.
1. Do the authors have any information about door (to produce candidate 1-3) including sex, age, mild-severe diseases etc? If so, this should be valuable information.
2. Figure 1: The authors showed the fold difference. Are they significant? They should perform the statics analysis and added the result in the Figure.
3. Figure 1B: The authors described XBB2.86 in the X-axis. But this should be BA.2.86. Please correct it.
4. Table 1: It looks like the authors added variant name (BA.5, XBB, BA.2.86 sublineages) on the top to distinguish the variants. However, “BA.5 sublineages” is on the XBB.15 and XBB1.9 and they are messed up. The authors should fix it.
5. Table 1: what “/” on the top (next to BA.5 sublineages) and “1” which show in the Table means? It’s confusing. Please clarify in the Table legend.
6. Table 3: What the red asterisk shown in XBB1.16 (RT/10) means? Please clarify in the Table legend.
Comments on the Quality of English LanguageMinor editing including spell check both in manuscript, Figures and Table are required.
Author Response
Comments 1: Do the authors have any information about donor (to produce candidate 1-3) including sex, age, mild-severe diseases etc? If so, this should be valuable information.
Response: Thanks for your suggestion. The information about donor has been supplied as Supplementary Table1.
Comments 2: Figure 1: The authors showed the fold difference. Are they significant? They should perform the statics analysis and added the result in the Figure.
Response: Thanks for your suggestion. Statistical analyses have been performed with Dunnett’s multiple comparison test. The results have been added in Figure 1 and Figure legend.
Comments 3: Figure 1B: The authors described XBB2.86 in the X-axis. But this should be BA.2.86. Please correct it.
Response: We are very sorry for our mistake. The “XBB.2.86” in the X-axis of Figure 1B has been corrected.
Comments 4: Table 1: It looks like the authors added variant name (BA.5, XBB, BA.2.86 sublineages) on the top to distinguish the variants. However, “BA.5 sublineages” is on the XBB.15 and XBB1.9 and they are messed up. The authors should fix it.
Response: We are very sorry for our incorrect writing. We have fixed the table.
Comments 5: Table 1: what “/” on the top (next to BA.5 sublineages) and “1” which show in the Table means? It’s confusing. Please clarify in the Table legend.
Response: We are very sorry for our negligence of explaining the “/” on the top and “1” in the Table. “/” on the top has been changed with “ancestral”. “1” in the Table 1 means the laboratory used the related virus to detect neutralizing antibody. We have been “1” changed to“√”.
Comments 6: Table 3: What the red asterisk shown in XBB1.16 (RT/10) means? Please clarify in the Table legend.
Response: We are very sorry for our mistake. The red asterisk in XBB1.16 (RT/10) did not have meaning, so it has been deleted.
Comments 7: Minor editing including spell check both in manuscript, Figures and Table are required.
Response: we are very sorry for our wrong spell in Figures and table. We have corrected them.
